# Risk factors of thromboembolic events in patients with scrub typhus

**Young Jae Ki[1], Sung Soo Kim[1], Jun-Won Seo[1], Da Young Kim[1], Na Ra Yun[1], Choon-Mee Kim**  **[2], Dong-Min Kim**  **[1]***

1 Department of Internal Medicine, College of Medicine, Chosun University, Gwangju, Republic of Korea,
2 Premedical Science, College of Medicine, Chosun University, Gwangju, Republic of Korea

* drongkim@chosun.ac.kr

## Abstract

### Background

Thromboembolic events are a well-recognized cause of in-hospital deaths of patients with infectious diseases. However, thromboembolic events in patients with scrub typhus, caused by *Orientia tsutsugamushi* have rarely been reported. This study aimed to assess risk factors associated with thromboembolic events in patients with scrub typhus.

### Methods

All 93 scrub typhus patients' diagnoses were confirmed serologically or by positive nested polymerase chain reaction (PCR). The clinical and laboratory findings from 12 scrub typhus patients with thromboembolic events and 81 scrub typhus patients with nonthromboembolic events were retrospectively studied. To determine the factors implicated in thromboembolic events, we performed multivariate logistic regression analysis using the six independent factors identified by the univariate analysis.

### Findings

The mean age of the patients in the thromboembolic group was 76.4 years (median, 76 years), and in nonthromboembolic group it was 64.6 years (median, 65 years) ($P<0.001$). Thromboembolic events were observed in 12 patients. These events included acute coronary syndrome (n = 5), acute limb ischemia (n = 4), ischemic stroke (n = 1), deep vein thrombosis combined with pulmonary thromboembolism (n = 1), and left common iliac artery aneurysm with a thrombus (n = 1). According to multivariate analysis, the following four factors were significantly associated with the thromboembolic events: 1) treatment with rifampin (OR = 57.63; $P$ = 0.039; CI 1.230–2700.27)., 2) Taguchi genotype (OR = 41.5; $P$ = 0.028; CI 1.5–1154.6), 3) atrial fibrillation (OR = 9.4; $P$ = 0.034; CI 1.2–74.0), and 4) age (OR = 1.1; $P$ = 0.046; CI 1.0–1.3).

### Conclusions

Our study suggests that clinicians should be cautious when managing patients with scrub typhus to avoid the development of thromboembolic events, especially in patients with risk

**Data Availability Statement:** All relevant data are within the paper and its Supporting Information files.

**Funding:** The study was supported by chosun university grant number: K207530001-1, The

funders had no role in study design, data collection and analysis, decision to publish, or preparation of the manuscript.

**Competing interests:** The authors have declared that no competing interests exist.

factors such as treatment with rifampin, Taguchi genotype, atrial fibrillation, and advanced age.

## Author summary

Scrub typhus, a disease caused by *Orientia tsutsugamushi*, is typically treated with antibiotics. It can sometimes lead to serious complications, including thromboembolic events. These events can be life-threatening but have been rarely reported due to under-suspicion in clinical practice, as their rarity often leads to them being overlooked. In this regard, our study suggests that clinicians should be cautious when managing patients with scrub typhus to prevent the development of thromboembolic events, especially in patients with risk factors such as treatment with rifampin, Taguchi genotype, atrial fibrillation, and advanced age.

## Introduction

Scrub typhus is an acute febrile illness that is characterized by focal or disseminated vasculitis [1]. Transmission occurs through the bite of a mite (chigger) infected with *Orientia tsutsugamushi* and is widely prevalent in Asia and the western Pacific regions [2]. The clinical manifestations of scrub typhus are usually nonspecific (e.g., fever, chills, headache, and myalgia) and are easily treated with antibiotics and conservative therapy. However, severe complications have been reported that have high morbidity and mortality rates, including acute respiratory distress syndrome, pneumonitis, renal failure, hepatic failure, meningoencephalitis, gastrointestinal bleeding, myocarditis, septic shock, and disseminated intravascular coagulation (DIC) [1–4]. Although venous and arterial thromboembolism are well-recognized causes of in-hospital death, they have rarely been reported in patients with scrub typhus [5–9]. Moreover, no studies have been conducted on the clinical features and factors associated with thromboembolism in patients with scrub typhus. Investigation of the risk factors is needed to aid early diagnosis and reduce the mortality rate of patients with scrub typhus with thromboembolic events.

12 patients with scrub typhus and thromboembolic events who were treated between 2005 and 2015 were included in this study. Here, we describe the thromboembolic events in patients with scrub typhus. This study also sought to evaluate the clinical and laboratory characteristics of these patients and analyze the risk factors associated with thromboembolism by comparing cases and controls as thromboembolic and nonthromboembolic groups, respectively.

## Methods

### Ethical statements

This study protocol was approved by the Research Ethics Committee of Chosun University Hospital (CHOSUN 2017-07-019-001). Since this was a retrospective study using secondary data, written informed consent was waived by the ethics committee. Patient data were anonymized prior to analysis to ensure privacy and confidentiality. No participants under the age of 18 were included in this study, and all methods were conducted in accordance with relevant guidelines and regulations.

## Study design and participants

We conducted a retrospective analysis of 12 patients with scrub typhus who experienced thromboembolic events between 2005 to 2015, as well as 81 patients who experienced no thromboembolic events between 2012 and 2015. During which, 154 hospitalized patients had a clinical and epidemiological diagnosis of scrub typhus, and 81 patients were confirmed serologically or by positive nested polymerase chain reaction (PCR) using buffy coats or eschar samples. All patients were treated at the Chosun University Hospital, South Korea.

## Procedures

The thromboembolic events were: 1) deep vein thrombosis (DVT) of the lower extremities, 2) pulmonary embolism, 3) acute coronary syndrome, 4) ischemic stroke, 5) peripheral arterial occlusion, and 6) other arterial thromboembolisms. Cases of deep vein thrombosis, pulmonary embolism, and other arterial thromboembolisms required confirmation by objective testing using spiral computed tomography (CT). Acute coronary syndrome consisted of any of the following: 1) ST-elevation acute myocardial infarction (ischemic chest pain, ST elevation on EKG, with or without increased cardiac enzymes), 2) non-ST-elevation myocardial infarction (ischemic chest pain with elevated cardiac enzymes, but no ST elevation), 3) unstable angina (ischemic chest pain without increased cardiac enzymes), or 4) the need for a coronary artery bypass graft or percutaneous vascular intervention for ischemic chest pain. Ischemic stroke required a clinical diagnosis using non-contrast or contrast CT scanning for brain imaging to confirm the absence of intracranial hemorrhage or non-vascular causes. Peripheral artery occlusion consisted of objectively confirmed lower or upper extremity embolisms or thromboembolisms.

The DIC score was calculated by using the platelet count, D-dimer levels, prothrombin time (PT), and fibrinogen level, in accordance with the algorithm for the diagnosis of overt DIC recommended by the DIC scientific subcommittee of the International Society for Thrombosis and Hemostasis (ISTH) [10]. We used the fibrinogen/C-reactive protein (CRP) ratio instead of the fibrinogen level to further elevate the diagnostic and prognostic power of the ISTH recommendation in the overt DIC template [11]. A score ≥5 was indicative of overt DIC, whereas a score <5 indicated non-overt DIC.

The diagnosis of scrub typhus was confirmed when the indirect immunofluorescent antibody assay (IFA) titer against *O. tsutsugamushi* increased four times or more in paired serum samples, or when a positive reaction was observed in a nested PCR test targeting the 56-kDa gene of *O.tsutsugamushi* [2,12].

Nucleotide primers were designed based on the nucleotide sequence of the gene encoding the 56-kDa antigen in a Gilliam serovariant of *O. tsutsugamushi*. Primers 34 (5'-TCAAGCTT ATTGCTAGTGCAATGTCTGC-3') and 55 (5'-AGGGATCCCTGCTGCTGTGCTTGCTG CG-3') were used in the first PCR, and nested PCR primers 10 (5'-GATCAAGCTTCCCTC AGCCTACTATAATGCC-3') and 11 (5'-CTAGGGATCCCGACAGATGCACTATTAGG C-3') were used in the second PCR amplification, generating a 483 bp fragment. The genotypes of the amplified samples were identified using BLAST on NCBI, and the sequences of the regions encoding the *O. tsutsugamushi* 56 kDa protein were analyzed using the Clustal X program, as described previously [13].

We performed IFA, a serologic test used by the Korea Disease Control and Prevention Agency (KDCA), which detects serum IgM and IgG in patients who respond to *O. tsutsugamushi* antigens. To conduct IFA according to the KDCA method, human sera were subjected to two-fold serial dilutions starting from 1:16 and then reacted with *O. tsutsugamushi* antigens [14].

## Statistical analysis

Statistical analyses were performed using SPSS for Windows, version 24·0 (SPSS; IBM Corp., Armonk, NY). Continuous data were not normally distributed, and a non-parametric test (Mann-Whitney U test) was used to compare the groups. Noncontinuous data were expressed as frequencies or fractions and compared using Pearson's chi-square test. To determine the factors associated with thromboembolic events, we performed multivariate logistic regression analysis using the independent variables shown by the univariate analysis to be associated with the occurrence of thromboembolic events. The model's fit was assessed using the Omnibus Tests, Model Summary, and Hosmer-Lemeshow Test, and we confirmed that the model was well-suited to the data. Additionally, multicollinearity was checked using the Variance Inflation Factor (VIF), and the results indicated no significant multicollinearity issues.

## Results

Thromboembolic events were observed in 12 patients with scrub typhus, including acute coronary syndrome (n = 5), acute limb ischemia (n = 6), ischemic stroke (n = 1), deep vein thrombosis combined with pulmonary thromboembolism (n = 1), and left common iliac artery aneurysm with a thrombus (n = 1). One patient died in the case group (1 of 12 patients). The clinical features of these patients are summarized in Table 1, and several graphical representations of the study results are shown in Fig 1.

**Table 1. Summary of clinical characteristics for thromboembolic events in scrub typhus patients.**

| Patient Age/Sex | Clinical symptoms | Rash/ Eschar | Genotype | Comorbidity | Clinical Diagnosis | Treatment for TE | Antibiotics | Death |
|---|---|---|---|---|---|---|---|---|
| 1/69/M | Abdominal pain/Rt. leg swelling/ Mental change | N/Y | Boryong | NA | DVT, PTE | Anticoagulation, IVC filter | R for 11d | N |
| 2/70/M | Lt. leg paresthesia | N/Y | Boryong | AFIB | Acute limb ischemia | Thrombectomy, Anti-PLT agent | D for 6d | N |
| 3/79/F | Both leg pain | Y/Y | Boryong | AFIB HTN | Acute limb ischemia | Thrombolysis & PTA | D for 5d | N |
| 4/75/M | Lt. side motor weakness | Y/Y | Taguchi | IHD HTN DM | Ischemic stroke | Anti-coagulation | R for 13d | N |
| 5/80/M | Both leg weakness | Y/N | Boryong | DM HTN DVT | Acute limb ischemia | Anti-PLT agent | D for 5d | N |
| 6/84/F | Both leg cyanotic change | Y/Y | Boryong | HTN AFIB CHF | Acute limb ischemia | Anti-PLT agent | D for 5d | Y |
| 7/89/F | General weakness/Chill/Myalgia | Y/Y | Boryong | NS | STEMI | Anti-PLT agent | R for 5d | N |
| 8/70/M | General weakness | Y/Y | Taguchi | HTN DM | Lt. CIA aneurysm with thrombus | Anti-PLT agent | D for 5d | N |
| 9/76/F | Chest discomfort | N/Y | NA | HTN | NSTEMI | Anti-PLT agent | D for 8d | N |
| 10/77/M | Dyspnea | N/Y | NA | DM | NSTEMI | Anti-PLT agent | D for 5d | N |
| 11/77/M | Fever/Chill/Myalgia | Y/Y | Boryong | AFIB | Unstable angina | Anti-PLT agent | R for 5d | N |
| 12/72/F | Fever | Y/Y | NA | DM HTN | NSTEMI | Anti-PLT agent | T for 7d | N |

D = doxycycline. R = rifampin. T = tigecycline. DVT = deep vein thrombosis. PTE = pulmonary thromboembolism. IVC = inferior vena cava. AFIB = atrial fibrillation. PLT = platelet. PTA = percutaneous transluminal angioplasty. IHD = ischemic heart disease. HTN = hypertension. DM = diabetes mellitus. CHF = congestive heart failure. STEMI = ST elevation myocardial infarction. NSTEMI = non ST elevation myocardial infarction. CIA = common iliac artery. N = No. Y = Yes. NA = not applicable

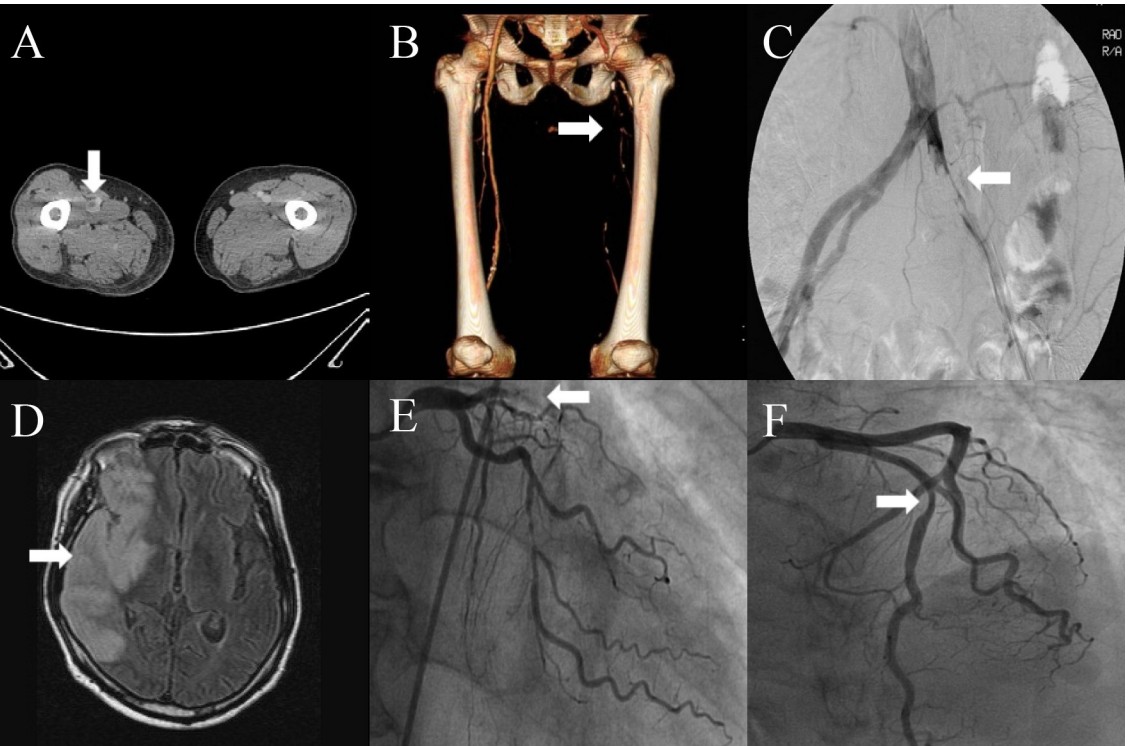

**Fig 1.** Computed venous tomography showed deep vein thrombosis in right femoral vein (A, Case 1, arrow). Peripheral CT angiography showed left common iliac artery total occlusion (B, Case 2, arrow). Peripheral angiography showed embolic total occlusion of left common iliac artery (C, Case 3, arrow). Brain CT scan indicated right middle cerebral artery infarction with mass effect and hemorrhagic transformation (D, Case 4, arrow). Coronary angiography showed a near total occlusion in the middle portion of left anterior descending coronary artery (E, Case 10, arrow). Coronary angiography showed moderate stenosis in the middle portion of left anterior descending coronary artery (F, Case 12, arrow).

The mean age of the patients in the thromboembolic group was 76.4 years (median, 76 years) and in the non-thromboembolic group it was 64.6 years (median, 65 years) ($P<0.001$). The frequency of atrial fibrillation was much higher in the thromboembolic group than in the nonthromboembolic group ($P = 0.015$). The frequency of rifampin administration was also much higher in the thromboembolic group ($P<0.001$). No significant differences were found in the fraction of men with genotypes of *O. tsutsugamushi*, except for those with the Taguchi genotype ($P = 0.063$). No significant differences were found in the frequencies of smoking, drinking, hypertension, diabetes mellitus, or cardiovascular diseases. There was no difference in the duration of symptoms before admission between the two patient groups, 6.75±5.4 days (median, 6 days) in the thromboembolic group and 6.23±3.6 days (median, 5 days) in the non-thromboembolic group. Overt DIC was observed in one patients in the nonthromboembolic group and was not observed in the thromboembolic group ($P = 0.598$). The occurrence of thromboembolic events was significantly correlated with higher APACHE II scores ($P<0.001$) and acute kidney injury ($P = 0.040$). White blood cell count, blood urea nitrogen, creatinine, creatinine kinase, CK-MB, and troponin-T were much higher, and PT was longer in the thromboembolic group (Table 2).

We used a univariate logistic regression model to test for correlations between the occurrence of thromboembolic events and the statistically and clinically presumed seven factors ($P<0.05$). The results are as follows: 1) treatment with rifampin (OR 40; $P<0.001$; CI 3.976–402.447), 2) Taguchi genotype (OR 7.9; $P = 0.005$; CI 0.999–62.445), 3) atrial fibrillation (OR

**Table 2. Summary of clinical characteristics and laboratory findings in the scrub typhus patients evaluated in a comparative study of thromboembolic events.**

| | Scrub typhus | | P-value[1] |
|---|---|---|---|
| | Non-thromboembolic group (n = 81) | Thromboembolic group (n = 12) | |
| Demographic characteristics | | | |
| Age(yrs) | 65(15–88) | 76(69–89) | <0·001 |
| Sex(Male) | 27(33.3%) | 6(50%) | 0.263 |
| Underlying diseases | | | |
| Smoking | 12(14·8%) | 2(16·7%) | 1·000 |
| Drinking | 8(9·9%) | 1(8·3%) | 1·000 |
| Hypertension | 27(33·3%) | 6(50%) | 0·335 |
| Diabetes mellitus | 15(18·5%) | 5(41·7%) | 0·124 |
| Atrial fibrillation | 5(6·2%) | 4(33·3%) | 0·015 |
| Cardiovascular disease | 3(3·7%) | 2(16·7%) | 0·123 |
| Laboratory finding | | | |
| WBC | 8250(1400–26330) | 10080(6080–26670) | 0·032 |
| Hb | 12·7(8·5–15·9) | 11·9(7·3–15·9) | 0·681 |
| Platelets | 144(34–410) | 121(69–313) | 0·305 |
| PT | 11·5(9·6–15·3) | 12·4(10·2–14·1) | 0·035 |
| aPTT | 31·1(23·1–383) | 31(25·8–42·6) | 0·710 |
| ESR | 17(2–83) | 20·5(2–62) | 0·713 |
| CRP | 9·26(0·05–24·1) | 9·83(3·49–21·9) | 0·333 |
| Glucose | 101(30–293·9) | 117·5(42–361) | 0·203 |
| Albumin | 3·35(2·37–4·20) | 3·33(2·78–4·05) | 0·801 |
| AST | 90·5(20·7–875·8) | 71·3(30·4–2499) | 0·443 |
| ALT | 71(23–393·7) | 44·3(13–691) | 0·121 |
| Total bilirubin | 0·76(0·29–3·59) | 0·88(0·48–1·6) | 0·159 |
| Na | 135(128–143) | 135(123–144) | 0·713 |
| K | 4(2·9–5·4) | 4·35(2·6–5·3) | 0·293 |
| Cl | 100(89–109) | 102.5(95–114) | 0·054 |
| BUN | 14·8(2·28–79·2) | 20·8(13·1–58·3) | 0·029 |
| Creatinine | 0·95(0·01–3·98) | 1·38(0·81–2·81) | 0·011 |
| CPK(n = 89) | 108(11–5462) | 415(35–42310) | 0·019 |
| CK-MB(n = 84) | 1·54(0·3–40·5) | 11·1(0·63–273) | <0·001 |
| Troponin-T(n = 84) | 0·011(0·001–0·28) | 0·028(0·01–0·86) | 0·009 |
| APACHE II scores | 7(0–17) | 13(6–18) | <0·001 |
| AKI | 12(14·8%) | 5(41·7%) | 0·040 |
| DIC | 1(1·2%) | 0 (0%) | 0·598 |
| Rifampin | 1(1·2%) | 4(33·3%) | <0·001 |
| Duration of illness before admission(day) | 6(0–20) | 5(1–17) | 0·596 |
| Death | 0(0%) | 1(8·3%) | 0·129 |
| Genotype(n = 78) | | | |
| Boryong | 59(85·5%) | 7(77·8%) | 0·622 |
| Karp | 8(11·6%) | 0(0%) | 0·586 |
| Taguchi | 2(2·9%) | 2(22·2%) | 0·063 |

1) Statistical significance test was done by Mann-Whitney U test

Values are mean ± SD. ESR = erythrocyte sedimentation rate. CRP = C-reactive protein. AST = aspartate aminotransferase. ALT = alanine aminotransferase.

LDH = Lactate dehydrogenase. CPK = Creatine Kinase. CK-MB = creatine kinase-MB.

**Table 3. Unadjusted and adjusted relative risk of thromboembolic events.**

| | Unadjusted Odds ratio | Adjusted | | P-value |
|---|---|---|---|---|
| | | Odds ratio | 95% CI | |
| Rifampin | 40·0 | 57·630 | 1·230–2700·27 | 0·039 |
| Taguchi genotype | 7·9 | 41·489 | 1·491–1154·60 | 0·028 |
| Atrial fibrillation | 7·6 | 9·385 | 1·184–73·986 | 0·034 |
| APACHE II scores | 1·385 | 1·252 | 0·973–1·611 | 0·080 |
| Age | 1·137 | 1·144 | 1·002–1·305 | 0.046 |

7.6; *P* = 0.008; CI 1.690–34.167), 4) acute kidney injury (OR 4.107; *P* = 0.033; CI 1.118–15.087), 5) Creatinine (OR 2.452; *P* = 0.024; CI 1.126–5.338), 6) APACHE II scores (OR 1.385; *P*<0.001; CI 1.158–1.658), 7) age (OR 1.13; *P* = 0.004; CI 1.042–1.240). Because creatinine level was included in the definition of acute kidney injury, six factors were included in the multivariate model of thromboembolic events. Because both backward and forward model selection methods produced similar results, data from the backward selection model are presented. Multivariate analysis demonstrated that the following four factors were significantly associated with thromboembolic events (Table 3): 1) treatment with rifampin (OR = 57.63; *P* = 0.039; CI 1.230–2700.27)., 2) Taguchi genotype (OR = 41.48; *P* = 0.028; CI 1.491–1154.60), 3) atrial fibrillation (OR = 9.358; *P* = 0.034; CI 1.184–73.986), and 4) age (OR = 1.14; *P* = 0.046; CI 1.002–1.305).

## Discussion

Venous and arterial thromboembolisms can be life-threatening complications among postoperative and/or immobilized medical patients [15]. Some studies have been conducted that explored the occurrence of DVT, stroke, and cardiac complications in patients who are critically ill or in the ICU [13–14,16]. One study recently surveyed the prevalence of venous and arterial thromboembolisms in patients with severe sepsis [17]. However, there are only a few reported cases of splenic infarcts, acute myocardial infarction, and cerebral infarction due to scrub typhus [5–8], and no study has been performed to evaluate the risk factors for thromboembolic events caused by scrub typhus.

Inflammation and other stressors, such as high blood pressure, can cause fibrous cap thinning, which leads to increased plaque instability due to endothelial damage. Eventually, atherosclerotic plaque erosion or rupture causes arterial thrombosis. The mechanism of venous thromboembolism has been explained by Virchow's triad, which consists of blood stasis, hypercoagulability, and endothelial damage. Endothelial damage significantly contributes to the development of arterial and venous thrombosis. The pathophysiology of scrub typhus is generalized vasculitis with subsequent vascular damage that involves multiple organs (lung, liver, kidney, heart, and brain), which are caused by multiplication of microorganisms in endothelial cells of microcirculation [18]. Liam *et al.* reported that acute infection is associated with a transient increase of the risk for vascular events, such as myocardial infarction and stroke [19]. Kim *et al.* reported that endothelial dysfunction by scrub typhus may affect the stability of atherosclerotic plaques [7]. Therefore, we can hypothesize that endothelial damage by scrub typhus infection, combined with systemic inflammation, may have the ability to affect thrombus formation.

In patients with severe sepsis, the most commonly observed thromboembolic events are ischemic stroke, acute coronary syndrome, and venous thromboembolism [17]. However, among the thromboembolic events in our study, acute coronary syndrome was the most

common (5 of 12), followed by acute limb ischemia (4 of 12), while ischemic stroke and venous thromboembolism were far less common. These results suggest that acute limb ischemia is more common in patients with scrub typhus than in those with severe sepsis. In contrast, the incidence of venous thromboembolism may be less common in patients with scrub typhus than in those with severe sepsis. These differences may occur because the clinical manifestations of venous thromboembolism are more obscure than those of other thromboembolic events, whereas the prominence of acute limb ischemia and acute coronary syndrome may be due to a clearer clinical presentation of these complications. Thus, the true prevalence of venous thromboembolism is probably higher than that measured in this study, and clinicians should pay special attention to the identification of thromboembolic events in the absence of evident clinical manifestations.

Traditional cardiovascular risk factors (hyperlipidemia, smoking, diabetes, hypertension, and obesity) are also associated with venous thromboembolism. A meta-analysis of 63,552 patients revealed that the relative risk of venous thromboembolism is 2.33 (95% CI 1.68–3.24) for obesity, 1.42 (95% CI 1.12–1.77) for diabetes, and 1.51 (95% CI 1.23–1.85) for hypertension [20]. In a large, population-based case-control study [Multiple Environmental and Genetic Assessment (MEGA) study], the relative risk of venous thromboembolism was determined to be 1.42 (95% CI 1.28–1.58) in smokers and 1.23 (95% CI 1.10–1.37) in former smokers, compared to the risk in individuals who had never smoked [21]. In our study, we did not evaluate patient BMI. Frequencies of smoking, hypertension, and diabetes mellitus were higher in the thromboembolic group than in the nonthromboembolic group. No statistically significant difference was found between the two groups, although this may be explained by the small sample size of the thromboembolic group.

Scrub typhus patients with thromboembolic events were generally older, had higher rates of atrial fibrillation, were more likely to have the Taguchi genotype, and were more likely to have treatment with rifampin. In the multivariate analysis, these four factors were found to be independent predictive variables for the occurrence of thromboembolic events. According to a previous study, most patients with scrub typhus in Korea are between 50–69 years of age, which can be explained by the increasing age of the farming population in rural villages [22]. In our study, the mean age of the nonthromboembolic group was 64.6 years (median, 65 years), whereas that of the thromboembolic group was 76.4 years (median, 76 years). This result is consistent with a previous study conducted on patients with severe sepsis [17]. Risk of both arterial and venous thromboembolism increase with age, and possible mechanisms include progressively increased plasma concentrations of coagulation factors (factors V, VII, VIII, IX, and fibrinogen), von Willebrand factor, and elevated inflammatory cytokine (interleukin-6, C-reactive protein) as age increases [23]. In this way, we should suspect thromboembolic events in older patients with scrub typhus who have definite or vaguely related symptoms.

Atrial fibrillation is the most common cardiac arrhythmia and is associated with substantial morbidity and mortality, frequently because of thromboembolic complications. A previous study concluded that new-onset atrial fibrillation in patients with severe sepsis was associated with an increased risk of stroke and hospital mortality [24]. Another study concluded that pre-existing atrial fibrillation was frequent in patients who were hospitalized with pneumonia, which is a marker of increased arterial thromboembolism and death [25]. In our study, four patients had atrial fibrillation upon admission; however, previous ECG data were unknown, so we could not determine if the conditions were new-onset or from pre-existing atrial fibrillation. However, we found that scrub typhus patients with atrial fibrillation had an increased risk of arterial and venous thromboembolic events compared to patients without atrial fibrillation. We can speculate that pathophysiological changes such as endothelial dysfunction and

systemic inflammation during scrub typhus infection may trigger thromboembolic events in patients with atrial fibrillation.

The dominant strains of scrub typhus are different in each endemic country. For example, the most common serotypes in Japan, Korea, and Guanzhou, China are Kawasaki, Boryong, and Karp, respectively. In our study, the Boryong genotype was dominant, followed by Karp genotype in nonthromboembolic group and Taguchi genotype in thromboembolic group. The Taguchi strain is one of the common *O. tsutsugamushi* strains in Korea, which is similar to the Taiwan-15 strain, with nucleotide sequence similarity reaching 99.37% [26]. The Taguchi genotype is a risk factor for thromboembolic events. Although several studies have been conducted on the differences in virulence between genotypes of *O. tsutsugamushi* in mice and on the clinical and laboratory features of scrub typhus involving patients with varying genotypes [27], no study has investigated the association between genotypes and complications in patients with scrub typhus. Consequently, further studies are needed and physicians should be aware of the possible development of thromboembolic events in scrub typhus patients with the Taguchi genotype.

Watt *et al*. reported that some strains of *O. tsutsugamushi* from northern Thailand are resistant to doxycycline and, in that case, rifampin seems to be a good alternative treatment option. Rifampin has also been used concurrently with doxycycline in patients with scrub typhus [28]. However, a case report of a patient with human immunodeficiency virus reported cerebral infarction and shock after re-administration of rifampin [26]. Rifampin can cause symptoms associated with liver toxicity, such as jaundice and increased liver enzymes, although adverse hematologic effects, such as venous thrombosis, may also occur [29]. Therefore, the possibility of a thromboembolic event caused by the administration of rifampin cannot be entirely ruled out, especially in patients suspected of having a coagulation disorder. However, the mechanism of rifampin-induced thrombosis requires further investigation.

This study had several limitations. First, it was a retrospective study conducted at a single institution. Second, we could not calculate the incidence of thromboembolic events because of the retrospective nature of the data of the admitted patients. Third, we could not fully assess other thromboembolic risk factors such as thrombophilia, trauma, surgery, cancer, and oral contraceptives. Finally, we did not know the exact date of onset of the thromboembolic events; rather, we used the date of diagnosis.

In conclusion, our study suggests that these four factors, namely treatment with rifampin, Taguchi genotype, presence of atrial fibrillation, and older age are possible risk factors for thromboembolic events in patients with scrub typhus. The presence of one or more of these four risk factors should alert clinicians to provide close monitoring and intensive care to prevent thromboembolic events.

## Author Contributions

**Conceptualization:** Young Jae Ki.

**Methodology:** Choon-Mee Kim.

**Supervision:** Sung Soo Kim, Jun-Won Seo, Da Young Kim, Na Ra Yun, Choon-Mee Kim, Dong-Min Kim.

**Writing – original draft:** Young Jae Ki, Dong-Min Kim.

**Writing – review & editing:** Young Jae Ki, Sung Soo Kim, Jun-Won Seo, Da Young Kim, Na Ra Yun, Choon-Mee Kim, Dong-Min Kim.

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
