## [Decision Letter · Decision Letter 0]

26 Feb 2024

Dear Dr. Kim,

Thank you very much for submitting your manuscript "Risk factors of thromboembolic events in patients with scrub typhus" for consideration at PLOS Neglected Tropical Diseases. As with all papers reviewed by the journal, your manuscript was reviewed by members of the editorial board and by several independent reviewers. In light of the reviews (below this email), we would like to invite the resubmission of a significantly-revised version that takes into account the reviewers' comments. 

We cannot make any decision about publication until we have seen the revised manuscript and your response to the reviewers' comments. Your revised manuscript is also likely to be sent to reviewers for further evaluation.

Sincerely,

Husain Poonawala

Academic Editor

Ana LTO Nascimento

Section Editor

Reviewer's Responses to Questions

**Key Review Criteria Required for Acceptance?**

**Methods**

-Are the objectives of the study clearly articulated with a clear testable hypothesis stated?

-Is the study design appropriate to address the stated objectives?

-Is the population clearly described and appropriate for the hypothesis being tested?

-Is the sample size sufficient to ensure adequate power to address the hypothesis being tested?

-Were correct statistical analysis used to support conclusions?

-Are there concerns about ethical or regulatory requirements being met?

Reviewer #1: The rationale behind the selection of the control population needs to be clarified.

Reviewer #2: The case-control design is appropriate for this retrospective study. 

My main comment with the method would be that the diagnosis of acute coronary syndrome is taken as a proxy for TE complication of scrub typhus even though the definitions used do not necessarily imply TE disease. This is particularly important as there were 5 patients with ACS out of 12 total TE patients. I note that coronary angiogram was performed in case 10 and 12. Was this performed on all 5 patients with ACS with TE confirmed? There are a number of causes of chest pain, raised cardiac enzymes and ECG changes including and not limited to myocarditis, pericarditis, coronary artery spasm, tachyarrhythmias and rate-related ischaemia, myocardial hypoperfusion due to septic shock etc. 

Indeed, myocarditis and pericarditis have been described as complications of acute scrub typhus infection.

Please clarify if echocardiographic findings, myoperfusion scan findings, coronary angiogram findings or CT coronary angiogram findings are available to support the diagnosis of TE disease.

Additionally, other factors that may increase or decrease the risk of TE disease could help improve the quality of data, in particular the presence of lines (midlines or PICCs), the administration of low molecular heparin (as VTE prophylaxis) or being established on anticoagulation (warfarin, DOACs). The last 2 factors may be protective while the presence of lines may increase the risk of VTE disease.

With regards to atrial fibrillation, it would be useful to distinguish fast poorly controlled AF (or very slow AF) with established persistent AF with acceptable rate. It is likely that in the study cohort of acutely unwell patients with scrub typhus, the label AF signifies the former rather than the latter. This is important as thromboembolic risk is likely to differ between these groups and fast AF (and slow AF if present) will also contribute to APACHEII scoring.

The data presented suggests to me that scrub typhus patients who are sicker are at higher risk of TE complications as signified by the statistically significant correlation with the APACHEII score. There is a risk of collinearity by including APACHEII scoring with other variables that contribute to the score in the multivariate logistic regression analysis (e.g. age, creatinine/AKI and potentially AF - if this is defined as fast AF or very slow AF). Please consider removing APACHEII score from the multivariate analysis to see if this change affects the results.

Patients were treated with doxycycline or rifampin with the latter being significantly correlated with TE complications. I am unfamiliar with the rationale/national guidelines in Korea for choosing one treatment over the other but in other settings, access to IV doxycycline is limited and patients who are more unwell and unable to tolerate oral intake may be given an IV antibiotic instead (such as rifampin). Is the rifampin correlation due to the drug itself or is it because the patients who received rifampin were more unwell? A quick way to check could be to compare the average APACHEII score between patients receiving doxycycline vs rifampin.

I may have missed this but please also add details on the method of genotyping for Ot used in this study.

**Results**

-Does the analysis presented match the analysis plan?

-Are the results clearly and completely presented?

-Are the figures (Tables, Images) of sufficient quality for clarity?

Reviewer #1: Some re-writing of the entire MS is needed to present the data more clearly. Too many unnecessary abrreviations in Table 1 - the entire names of the analytes can be written in column 1 as there is enough space for this.

Reviewer #2: The analysis presented matches the plan laid out in the method.

I suggest the addition of a table (either within the main text or as a supplement) which shows the results of the univariate logistic regression analysis with OR with 95% CI and p values. This will make it clear which variables were then selected for the multivariate analysis.

**Conclusions**

-Are the conclusions supported by the data presented?

-Are the limitations of analysis clearly described?

-Do the authors discuss how these data can be helpful to advance our understanding of the topic under study?

-Is public health relevance addressed?

Reviewer #1: The authors should more clearly state how clinicians can (or cannot) make use of the identified risk factors when taking care of a patient with scrub typhus.

Reviewer #2: I have highlighted queries above.

If it is not common practice already, it could be suggested that prophylaxis for thromboembolic disease (e.g. with low molecular weight heparin) be used in patients acutely unwell with scrub typhus if there are no contra-indications or the patient is already not on anticoagulation. If it is already common practice, this should be emphasised.

**Editorial and Data Presentation Modifications?**

Reviewer #1: (No Response)

Reviewer #2: As above.

**Summary and General Comments**

Reviewer #1: The objective and findings of the study are of interest but need to be presented more clearly.

Reviewer #2: The authors present the results of a retrospective case control study on thromboembolic complications in patients with scrub typhus. This is a useful study to investigate a clinical complication of the disease which has not been extensively studied in the past. There are some limitations including the retrospective design, the limited number of thromboembolic complications noted and some weaknesses in the data collected and analysis method.

I have outlined these points above and hope the authors will be able to clarify +/- provide additional data and analysis to improve the overall quality of results and strengths of conclusions.

Line 48 - please add "through the bite of the larval stage of a mite (chigger)...". The nymphal and adult stages do not feed on vertebrate hosts.

PLOS authors have the option to publish the peer review history of their article (what does this mean?). If published, this will include your full peer review and any attached files.

Reviewer #1: No

Reviewer #2: No
---

## [Decision Letter · Decision Letter 1]

30 Jul 2024

Dear Dr. Kim,

Thank you very much for submitting your manuscript "Risk factors of thromboembolic events in patients with scrub typhus" for consideration at PLOS Neglected Tropical Diseases. As with all papers reviewed by the journal, your manuscript was reviewed by members of the editorial board and by several independent reviewers. The reviewers appreciated the attention to an important topic. Based on the reviews, we are likely to accept this manuscript for publication, providing that you modify the manuscript according to the review recommendations. 

Sincerely,

Husain Poonawala

Academic Editor

Ana LTO Nascimento

Section Editor

Reviewer's Responses to Questions

**Key Review Criteria Required for Acceptance?**

**Methods**

-Are the objectives of the study clearly articulated with a clear testable hypothesis stated?

-Is the study design appropriate to address the stated objectives?

-Is the population clearly described and appropriate for the hypothesis being tested?

-Is the sample size sufficient to ensure adequate power to address the hypothesis being tested?

-Were correct statistical analysis used to support conclusions?

-Are there concerns about ethical or regulatory requirements being met?

Reviewer #3: The study's objectives are clearly articulated, with a well-defined and testable hypothesis. The chosen study design, a case-control format, is appropriate for addressing the stated objectives. The population is clearly described and aligns well with the hypothesis under investigation. 

The followings need further declaration and correction:

1. The significant p-value of univariate analysis was not being described. 

2. Did multicollinearity and interaction terms were checked? Did Hosmer Lemeshow test and classification table were applied to check the model fitness? Kindly declare this in the statistical analyses.

3. Line 105 - double 'were' noted in the sentence - please check the context of the sentence

4. Table 2 title is positioned wrongly -below the table 

5. Reference no 14 and 29 are in different format.

**Results**

-Does the analysis presented match the analysis plan?

-Are the results clearly and completely presented?

-Are the figures (Tables, Images) of sufficient quality for clarity?

Reviewer #3: The analysis presented aligns well with the predefined analysis plan, ensuring methodological consistency. The results are clearly and comprehensively presented, facilitating an understanding of the study's outcomes. Additionally, the quality if tables are sufficient to ensure clarity, effectively supporting the findings of the research.

**Conclusions**

-Are the conclusions supported by the data presented?

-Are the limitations of analysis clearly described?

-Do the authors discuss how these data can be helpful to advance our understanding of the topic under study?

-Is public health relevance addressed?

Reviewer #3: The conclusions drawn in the study were supported by the data presented. The authors have clearly described the limitations of the analysis, providing a comprehensive understanding of the potential constraints and biases inherent in the study. The discussion highlights how the data can advance our understanding of risks of thromboembolic events in scrub typhus.

**Editorial and Data Presentation Modifications?**

Reviewer #3: Minor Revision

**Summary and General Comments**

Reviewer #3: This manuscript offers a valuable assessment of the risk of thromboembolic events in scrub typhus, a significant yet rare complication of the disease. The study's strength lies in its focus on a rarely discussed topic. However the main limitation is the small sample size within the case group.

PLOS authors have the option to publish the peer review history of their article (what does this mean?). If published, this will include your full peer review and any attached files.

Reviewer #3: No

Figure Files:

Data Requirements:

Reproducibility:

References

---

## [Decision Letter · Decision Letter 2]

9 Sep 2024

Dear Dr. Kim,

We are pleased to inform you that your manuscript 'Risk factors of thromboembolic events in patients with scrub typhus' has been provisionally accepted for publication in PLOS Neglected Tropical Diseases.

Best regards,

Husain Poonawala

Academic Editor

Ana LTO Nascimento

Section Editor

Reviewer's Responses to Questions

**Key Review Criteria Required for Acceptance?**

**Methods**

-Are the objectives of the study clearly articulated with a clear testable hypothesis stated?

-Is the study design appropriate to address the stated objectives?

-Is the population clearly described and appropriate for the hypothesis being tested?

-Is the sample size sufficient to ensure adequate power to address the hypothesis being tested?

-Were correct statistical analysis used to support conclusions?

-Are there concerns about ethical or regulatory requirements being met?

Reviewer #3: The objectives were clearly articulated.

The study design was appropriate for the stated objectives.

The population was clearly described and appropriate for the hypothesis tested.

**Results**

-Does the analysis presented match the analysis plan?

-Are the results clearly and completely presented?

-Are the figures (Tables, Images) of sufficient quality for clarity?

Reviewer #3: The analysis matched the analysis plan.

The results were clearly and completely presented.

The tables were sufficient in quality for clarity.

**Conclusions**

-Are the conclusions supported by the data presented?

-Are the limitations of analysis clearly described?

-Do the authors discuss how these data can be helpful to advance our understanding of the topic under study?

-Is public health relevance addressed?

Reviewer #3: The conclusions were supported by the data presented.

The limitations of the analysis were clearly defined.

**Editorial and Data Presentation Modifications?**

Reviewer #3: Accept

**Summary and General Comments**

Reviewer #3: The authors have made relevant corrections and modifications in the article.

PLOS authors have the option to publish the peer review history of their article (what does this mean?). If published, this will include your full peer review and any attached files.

Reviewer #3: No

---

## [Editor Report · Acceptance letter]

26 Sep 2024

Dear Dr. Kim,

We are delighted to inform you that your manuscript, "Risk factors of thromboembolic events in patients with scrub typhus," has been formally accepted for publication in PLOS Neglected Tropical Diseases.

Best regards,

Shaden Kamhawi

co-Editor-in-Chief

Paul Brindley

co-Editor-in-Chief
